# Digital Twin-Driven Tool Condition Monitoring for the Milling Process

**DOI:** 10.3390/s23125431

**Published:** 2023-06-08

**Authors:** Sriraamshanjiev Natarajan, Mohanraj Thangamuthu, Sakthivel Gnanasekaran, Jegadeeshwaran Rakkiyannan

**Affiliations:** 1Department of Mechanical Engineering, Amrita School of Engineering, Amrita Vishwa Vidyapeetham, Coimbatore 641112, India; cb.en.p2edn20032@cb.students.amrita.edu; 2Centre for Automation, School of Mechanical Engineering, Vellore Institute of Technology (VIT), Chennai 600127, India; sakthivel.g@vit.ac.in (S.G.); jegadeeshwaran.r@vit.ac.in (J.R.)

**Keywords:** condition monitoring, milling process, sound, vibration signals, machine learning, digital twins

## Abstract

Exact observing and forecasting tool conditions fundamentally affect cutting execution, bringing further developed workpiece machining accuracy and lower machining costs. Because of the unpredictability and time-differing nature of the cutting system, existing methodologies cannot achieve ideal oversight progressively. A technique dependent on Digital Twins (DT) is proposed to accomplish extraordinary accuracy in checking and anticipating tool conditions. This technique builds up a balanced virtual instrument framework that matches entirely with the physical system. Collecting data from the physical system (Milling Machine) is initialized, and sensory data collection is carried out. The National Instruments data acquisition system captures vibration data through a uni-axial accelerometer, and a USB-based microphone sensor acquires the sound signals. The data are trained with different Machine Learning (ML) classification-based algorithms. The prediction accuracy is calculated with the help of a confusion matrix with the highest accuracy of 91% through a Probabilistic Neural Network (PNN). This result has been mapped by extracting the statistical features of the vibrational data. Testing has been performed with the trained model to validate the model’s accuracy. Later, the modeling of the DT is initiated using MATLAB-Simulink. This model has been created under the data-driven approach. The physical–virtual balance of the DT model is acknowledged utilizing the advances, taking into consideration the detailed planning of the constant state of the tool’s condition. The tool condition monitoring system through the DT model is deployed through the machine learning technique. The DT model can predict the different tool conditions based on sensory data.

## 1. Introduction

Machining is essential in the manufacturing sector because it observes time and cost. The tool’s life is proportional to its machining parameters and work environment. Process monitoring is becoming more popular in manufacturing to improve machine tool usage efficiency and productivity while minimizing quality losses. A total of 75% of all production downtime occurs in relation to tooltip wear during milling. According to research, the maintenance of machine tools’ essential parts accounts for 12% of the cost of manufacturing [1]. Tool condition monitoring (TCM) is taken care of to ensure the tool’s life in the long run. Proper process monitoring is crucial to work on the viability of the machining process through the metal-cutting strategy [2]. To forecast and maximize the life of the essential components of machine tools, the systems used for this purpose can monitor their working conditions to ensure the overall effectiveness of production. Direct and indirect tool condition monitoring are the two available methods. The dividing line between the approaches is not clearly defined, and many techniques can be classified as belonging to either category [3]. The vibration, acoustic emission, cutting force, and spindle current signals have a significant role in condition monitoring [4].

Milling is a technique for removing material by a series of tiny incisions. Machining can be accomplished by using a tool with several teeth, spinning the tool more quickly, or carefully forcing the material through the tool; most frequently, a combination of these three techniques is utilized. In this processing environment, the check on a tool’s condition (TC) has been observed to be very critical. This technique increases manufacturing intelligence, networking, and digitalization [5]. This move puts fault diagnosis and prognosis under pressure and calls for the new system to analyze how the operating environment affects prediction and improves the accuracy of prediction findings. Studies on fault diagnosis and prognosis have been conducted during the last few decades [6]. The tool replacement time for the machining process was analyzed and optimized [7].

The study of condition monitoring, analysis, and forecasting using the Digital Twins approach methodology is only getting established. A digital twin is a virtual model that accurately reflects a physical object. The object being studied is cutting tools and their tool wear, outfitted with sound and vibration signals to monitor the tool condition. There are currently the following issues: (A) the model’s foundation—there is a significant issue with the influence of the working environment or conditions of the device system; (B) the information storage capacity—there is not enough space to retain the acquired data; (C) the system’s feedback loop [6]. However, the industrial environment typically does not match vision-based procedures due to several sources of interference, such as cutting fluid and material chips. Therefore, even if computer vision systems perform well in a lab setting, they must undergo significant adjustments before being implemented in an existing machining line. 

TCM is established on a data-driven methodology, where the wear assessment is derived from incoming sensor data without reference to the physical principles governing the milling process. Model-based techniques relate to analyzing tool wear based on tool life equations derived from fundamental physical laws of the material removal process, as opposed to the data-driven approach [8]. Deep learning (DL)-based TCM techniques have received much attention, particularly with advanced data and graphics processing unit (GPU) technology. Artificial neural networks (ANN) represent learning as the foundation for deep learning [9]. IoT-based design, modeling, and condition monitoring have also been performed [10].

The DT innovation can recreate processes to gather information and guess how they will perform. A DT model creates simulations that rely on precise data to predict the quality of the product. DT opts for Industry 4.0 to refine the outcome further. Because of advancements in AI and factors such as important information, these virtual models have become standard in modern design to guide development and enhance execution. They are improving a numerical model that re-enacts the first starts, with experts in applied math or information science analyzing the physical and functional information of an actual item or framework [11]. DT can be as convoluted or essential as it wants, with the information utilized choosing how intently the model emulates the accurate rendition. DT can be used to model and criticize the item as it is being made, or it tends to be utilized as a model all by itself to re-enact what may happen when the genuine adaptation is assembled [11]. 

Advanced twins can be utilized for various purposes, such as testing a model or configuration, deciding and observing lifecycles, and evaluating how an item or process would perform under different conditions. Information is accumulated, and computational models are made to test the DT idea. This could incorporate a continuous point of interaction between the computerized model and an existing system for sending and getting information. DT is created by collecting information about the system, modeling the system, and connecting the twins with the physical system. Data collection requires information about the physical asset or cycle to create a virtual model that can reproduce. The data could be connected with a system’s lifecycle and incorporate plan prerequisites, fabricating techniques, or designing information.

To the authors’ knowledge, the DT technique for tool condition monitoring in the end-milling of Inconel 625 using vibration and sound signals was rarely found in the literature. This study examines the development of a DT model for end-milling tool condition monitoring with various machine learning techniques. In contrast to previous methods, the model was developed using machine learning techniques such as PNN, KNN, SVM, NB, and RF, considering the statistical characteristics of vibrational signals and sound data. The accuracy of several ML models’ predictions is evaluated using the confusion matrix. The validation test employs the DT model.

## 2. Materials and Method 

### 2.1. Selection of Workpiece

The workpiece material is Inconel 625, prized for its enhanced strength, superior fabric ability, and consumption obstruction. The cryogenic to 982 °C temperature range is significant. It is a profoundly severe material and challenging to process [12]. It is used in the automotive, maritime, and aviation industries. Its substance make-up is recorded by Nickel (58% least), trailed by chromium, and trace quantities of Manganese, Silicon, Aluminum, Titanium, Niobium, Iron, Tantalum, and Cobalt.

### 2.2. Cutting Tool

Even while cutting through a hardened workpiece, the coating prevents tool life failure in Titanium Carbide tools due to its exceptional heat and oxidation resistance. The tool can perform precision machining because of the cemented carbide’s excellent wear resistance. The cutting tools used for the experiment are shown in Figure 1.

### 2.3. Milling Operation

The Bharat Fritz Werner Gaurav BMV 35T12 milling machine was used for the CNC milling testing, and Figure 2 depicts the experimental setup. The cutting speed, feed, and depth interactions impact the milling stability. Therefore, selecting the optimal set of machining settings for the specific combination of cutting tool shapes may be wise to have vibration-free machining. The cutting parameters were fixed as constant to analyze the influence of tool conditions on the sound and vibration signals. The machining details and the ideal machining conditions for the experiment are listed in Table 1. With the mentioned machining parameters, the tools are classified into five categories: fresh tool, single-tip broken tool, two-tip broken tool, three-tip broken tool, and four-tip broken tool. With these classifications, vibrational and sound signals were captured. The sensors must be placed appropriately to acquire the signal accurately. 

### 2.4. Data Acquisition

An end-milling process was carried out on the workpiece using Titanium Carbide inserts in a CNC machine. A microphone sensor was mounted on the spindle housing, and an accelerometer was utilized to track the vibration signals, as shown in Figure 2. Vibration signals were measured with the NI 9234 sound and vibration measurement DAQ, with a sample rate of 20 K samples per second. The National Instruments DAQ system collected and stored analog signals [12]. Furthermore, the software Audacity managed the audio signal.

In most cases, the vibration in ‘y’ is more dominant than the other two axes. On the other hand, the literature suggests that vibration in the ‘z’ axis is dominant and most sensitive to wear [13,14,15] and related to the depth of cut value. In this work, Inconel 625 was machined with a depth of cut of 0.15 mm, and the vibration was measured along the ‘z’ axis.

Figure 3 shows the vibration and sound signals for the new and four-tip broken tools. The vibration signal for the new tool (0.005 g) is much lower than the four-tip broken tool (0.01 - 0.05 g). Similarly, the sound signals for the four-tip broken tool are also very high compared to the new tool.

### 2.5. Feature Extraction

Finding the most critical parameter with feature extraction is extracting meaningful information and reducing the amount of data [16]. Features collected from the obtained signals were fed as input to the ML algorithm to investigate the tool condition. A signal processing method with an acceptable temporal resolution is needed to predict when the tool will fail [12] correctly. Large-frequency vibration signals are produced during four-tip broken tool machining and can be detected with the proper temporal resolution. During the feature extraction procedure, the most acceptable properties highly correlated with tool wear and not inflated by cutting conditions were extracted [4]. Statistical characteristics were retrieved from the vibration signal, including mean, root means square, kurtosis, skewness, standard deviation, variance, median, summation, and mode. From the microphone, the different cutting conditions’ sound signals were captured. The software was used to transform the analog sound signals the microphone had acquired into digital signals.

### 2.6. Classifiers

A consistent classifier must be developed to illustrate the relationship between tool wear and sensory input. A model that attempts to derive conclusions from specific observable values is known as a categorization model. A categorization model seeks to estimate the output value for a given input. Typically, a dataset is labeled with these results [17]. Machine learning approaches are employed for categorization without sacrificing generalization ability due to the complexity of milling and the rise in uncertainty around tool wear [18]. 

A Probabilistic Neural Network (PNN) solves pattern recognition and classification issues. The probability distribution function (PDF) of each class is approximated by the PNN technique using a Parzen window and a non-parametric function. After calculating the class probability of new input data using the PDF of each class, Bayes’ rule is used to allocate the data to the class with the best posterior probability [19]. This strategy lessens the possibility of misclassification. This ANN was created using a Bayesian network and the statistical method known as kernel Fisher discriminant analysis [20]. 

A Support Vector Machine (SVM) may be applied to classification and regression issues. In any case, it is widely used with machine learning to solve categorization difficulties. SVM computation aims to identify the best line or option limit for classifying the n-layered space so that more information foci may be quickly added and placed in the appropriate category afterward. A hyperplane is referred to as the best option limit. The extreme points and vectors that comprise the hyperplane are chosen using SVM. The purpose of the computation, known as a Support Vector Machine, is to locate the plane with the most significant margin between the data points from the two classes. Increasing the limit distance adds more support, making it easier to group the following pieces. Decision boundaries, known as hyperplanes, help characterize information. Elements on each side of the hyperplane can be given a variety of characterizations. The quantity of features affects the hyperplane’s size as well. 

One of the most fundamental machine learning algorithms, the K-Nearest Neighbor (KNN) approach, relies on the supervised learning method. The KNN technique categorizes alternative information sources for how well they match the most recent information while preserving generally available information. The KNN approach finds the new case in the characterization commonly like the current classes and guesses that the new case/data are similar to existing occurrences. This suggests that new data may be promptly categorized into a distinct class with the KNN approach. Although the KNN technique may be used for characterization and relapse, it is most frequently employed for arrangement assignments. Since the KNN computation is non-parametric, it has no reason to suspect any information is concealed [11]. It is sometimes referred to as an apathetic student calculation since it maintains the dataset and takes action when it is time to order it, despite not immediately benefiting from the preparation. During the preparation stage, the KNN calculation effectively preserves the dataset. It describes newly received data in a class comparable to the new data.

Naive Bayes (NB), the ideology of this classifier, assumes that one trait’s occurrence has no bearing on the occurrence of other traits. It is categorized as naïve. For instance, a red, spherical, sweet fruit is recognized as an apple based on its color, shape, and flavor. Each attribute, therefore, works independently to aid in identifying the object as an apple. It is called the Bayes principle since it is based on Bayes’ Theorem. Based on the Bayes theorem, the NB methodology is a supervised learning method for classification problems. It primarily makes use of a considerable preparation set for characterization.

A component of the supervised learning method is Random Forest (RF). It could apply to ML problems that need classification and regression. A component of it is outfit learning, a technique for merging a few classifiers to overcome testing issues and enhance model performance [21]. By averaging many decision trees applied to various subsets of the available data, RF is a classifier that raises the expected accuracy of the dataset. It employs predictions from each tree rather than just one, forecasting the outcome based on the votes of the majority of projections [22].

## 3. TCM—A Digital Twins Approach

The DT innovation can recreate processes to gather information and guess how they will perform [23]. “Digital twins” originally referred to reflecting products and twining them in virtual spaces. When the decision was made to call the idea the DT, NASA Technology Roadmaps had already gained some notoriety in the aerospace industry [24]. The DT’s capacity to offer various information uniformly is a crucial feature. In addition to being made up of pure data, digital twins also contain algorithms that characterize their real-world counterparts and determine what should be done in the production system based on this processed data [24]. These virtual models have become a norm in current design to drive development and improve execution because of progressions in AI and components such as extensive information.

One can assist with working on key mechanical patterns, forestall expensive breakdowns in actual articles, and test cycles and administrations using unrivaled logical, checking, and creative abilities. A DT model is the improvement of a numerical model that re-enacts the first starts, with experts in applied math or information science analyzing the physical and functional information of an actual item or framework [23]. The makers of DT verify that the DT model might get criticism from sensors that gather information from this present reality adaptation. This permits the advanced variant to impersonate and copy what is going on with the first form, continuously considering the assortment of information on execution and possible issues. DT can be as convoluted or essential as they want it to be, with the amount of information utilized choosing how intently the model emulates the accurate rendition. The twins can be utilized in relation to a model to criticize the item as it is being made, or it tends to be utilized as a model by itself to re-enact what may happen when the genuine adaptation is assembled [25].

Advanced twins can be utilized for various purposes, such as testing a model or configuration, deciding and observing lifecycles, and evaluating how an item or process would perform under different conditions. Information is accumulated, and computational models are made to test the DT idea. This could incorporate a continuous point of interaction between the computerized model and an actual article for sending and getting criticism and information [23]. A DT requires information about the item or cycle to create a virtual model that can reproduce the practices or conditions of a simple thing or methodology. A digital model represents a physical thing, either already in existence or intended, that does not include any automatic data interchange between the real and digital objects [24]. Instead, the digital representation may have a more detailed description of the real thing. These models might represent a physical entity that does not use automatic data integration, including simulation models of future factories, mathematical models of novel products, or other physical things. Such models may still be created using digital data from existing physical systems, but all data transfer is performed manually [24]. As a result, the physical object’s state cannot directly affect the digital object and vice versa. 

When the information has been gathered, it may foster computational scientific models to demonstrate working effects, expect states such as exhaustion, and anticipate practices. Designing re-enactments, physical science, science, measurements, AI, computerized reasoning, business rationale, and destinations can be generally used to endorse activities [25]. These models can be displayed utilizing 3D portrayals and increased reality demonstrations to work on human comprehension of the information. Advanced twins’ revelations can be connected to create an outline, for example, taking hardware twins’ discoveries and putting them into a creation-line twins model, illuminating an industrial facility scale DT [11]. Advanced connected twins can empower brilliant modern applications for genuine functional developments and upgrades [26]. DT can be developed with a physics-based model and data-driven approach.

### 3.1. DT—Physics-Based Model

A “3D information twin model” could potentially refer to a digital representation or simulation of a three-dimensional object, system, or environment that captures and incorporates various information related to that object or system. This information could include geometric data, spatial relationships, material properties, textures, colors, and possibly dynamic aspects such as motion or behavior.

Such a model could be used in various fields, including computer graphics, virtual reality, augmented reality, computer-aided design (CAD), engineering simulations, architectural visualizations, and more. It would aim to provide an accurate and detailed representation of the physical entity in a digital form, enabling analysis, visualization, interaction, or other forms of manipulation. For the milling process, creation of a 3D model is challenging. This work uses the data-driven approach to develop the DT model through machine learning algorithms.

### 3.2. DT—Data-Driven Approach

The “Tool Twin Model” is a concept that refers to the digital representation or virtual counterpart of a physical tool or machine. It is commonly used in the context of Industry 4.0 and the Internet of Things (IoT) to enhance manufacturing processes and enable advanced functionalities. From the perspective of the mechanism, the establishment of a tool twin model involves several steps:➢Data Acquisition: The first step is to collect data about the physical tool or machine. Data acquisition can include information about its geometry, components, sensors, actuators, control systems, and operating parameters. Data acquisition methods may involve manual measurements, sensor integration, or leveraging existing CAD or digital design data.➢Data Integration: The acquired data need to be integrated into a unified digital representation. This involves combining the data with other relevant information. The data integration ensures that the digital model accurately represents the physical tool. Here, the data are integrated with different tool conditions.➢Model Development: Once the data are integrated, a machine learning model is created based on the acquired information. This model aims to capture the tool condition. Various machine learning algorithms are used to develop the model. The model which has the highest classification accuracy is used for DT. Here, the DT model is developed using the MATLAB/ Simulink toolbox.➢Calibration and Validation: The developed Simulink model must be calibrated and validated to ensure accuracy. This involves comparing the behavior of the digital model with real-world measurements or performance data obtained from the physical tool. Calibration may involve adjusting model parameters or refining the model structure to achieve better alignment with the physical system.➢Integration and Data Exchange: For real-time application, the developed model has to be integrated with a CNC machine to monitor the tool condition. This enables real-time monitoring, data exchange, and synchronization between the digital twin and its physical counterpart. Sensor data from the physical tool can be used to update and refine the digital model. In contrast, the digital twin can provide insights, predictions, or optimization suggestions to enhance the operation and maintenance of the physical tool.

Overall, establishing a tool twin model from the perspective of the mechanism involves acquiring, integrating, modeling, calibrating, and validating data to create a digital representation that closely resembles the physical tool or machine. This enables advanced analysis, optimization, predictive maintenance, and other value-added functionalities in Industry 4.0 and smart manufacturing.

With DT technology, an approach toward TCM has been charted in Figure 4. The optimal process parameter was selected, and the machining process was conducted with different simulated tool conditions. During the machining process, vibration and sound signals were measured. The statistical features, namely mean, root means square, kurtosis, skewness, standard deviation, variance, median, summation, and mode, were extracted from the vibration and sound signals. Then, the extracted features were given as input to the machine learning algorithms. The DT model was developed using the MATLAB/Simulink model with the machine learning algorithm. The developed DT model was verified with the testing datasets.

## 4. Results and Discussion

Initializing the ML involves running the DT, created by MATLAB machine learning technique (.m file, a graphics user interfaced file). The two options are ML analysis and data visualization. ML analysis contains a combination of different machine learning algorithms and their results. ML analysis imports the data from the current working directory: new.xlsx, one_tip_1.xlsx, two_tip_2.xlsx, three_tip_3.xlsx, four_tip_4.xlsx. The data include up to 339 rows that contain numerical information—after that follows having NaN values, adding the prediction class to the dataset, merging the prediction class to the existing dataset, and converting the table into an array concerning the size four_tip_4.xlsx file information. Because of the values in the fifth condition, four_tip_4.xlsx contains significantly less value when compared with the other four datasets.

They assign the independent feature and prediction class data as feature data and target class variables, respectively, and later split the data into training and testing. Of the total data, 75% will be allocated for training, and 25% for testing along with data, cross-validation, and the stratify methods enabled. The different ML classifiers (PNN, SVM, KNN, NB, and RF) and the corresponding template pass the training data to the classifier to train the corresponding model. Once the model is trained, the next step is to predict the trained model based on the test data. The classification accuracy can be identified using the confusion matrix.

When considering the robustness of the method used to establish a tool twin model, factors such as model sensitivity and accuracy play a vital role [27,28]. The accuracy value is calculated by comparing the actual and the predicted results. Those values will be visually represented by using the confusion matrix with summary data. The fresh, single-tip broken, two-tip broken, three-tip broken, and four-tip broken tools are labeled on the rows and columns of the 5 × 5 matrix. Cell (1, 1) indicates a new tool, (2, 2) indicates a single-tip broken tool, (3, 3) indicates a two-tip broken tool, (4, 4) indicates the three-tip broken tool, and (5, 5) indicates to the four-tip broken tool. This flow of order is followed in the rest of the algorithms too.

The confusion matrix obtained from PNN is shown in Figure 5. The algorithm yields a result of 87.6% of accuracy for the new tool, followed by the overall accuracy of 91%. The precision true positive rate (TPR), false negative rate (FNR), and false discovery rate are mentioned in the confusion matrix. The KNN algorithm’s confusion matrix is presented in Figure 6. The accuracy of 87.8%, 92.6%, 73.9%, 100%, and 75.6% is mapped for the five conditions of the tool, respectively. The KNN algorithm has an overall classification accuracy of 86%.

The confusion matrix for the SVM algorithm is shown in Figure 7. The algorithm comes up with a result of 75.75%, 94.0%, 82.9%, 100%, and 81.6% accuracy for the corresponding TC. The overall accuracy of the SVM method for the input dataset is 87%.

The NB algorithm’s confusion matrix is shown in Figure 8. The accuracy of 65.8%, 88.4%, 80.4%, 98.8%, and 86.4% is mapped for the five conditions of the tool, respectively. The NB algorithm has a total accuracy of 83% for the input dataset. The confusion matrix for the RF algorithm can be seen in Figure 9. The algorithm comes up with a result of 85.9%, 94.28%, 86.4%, 100%, and 81.4% of accuracy for the corresponding TC. Therefore, the total accuracy of the RF method for the input dataset is observed as 90%. Each classifier’s overall accuracy is charted in Table 2.

### 4.1. Accuracy Test

The resulting confusion matrix from the five ML algorithms gives the predicted accuracy of the study, wherein the PNN algorithm obtains the maximum accuracy of 91%. Furthermore, the SVM, KNN, NB, and RF were mapped accordingly with 87%, 86%, 83%, and 90% accuracy. The ML analysis was performed with the different TC to validate the observed results, and the findings were good. The benchmark for the accuracy validation from the paper findings is in Table 3.

### 4.2. Validation Test on the Digital Twins Model

The randomly chosen testing dataset is allowed to validate the developed DT model. The chosen values of the dataset are mapped with the proper tool condition. Moreover, the ML and audio analysis results in good mappings sample validated model are shown in Figure 10. The developed DT model predicts the tool condition with better accuracy, indicating that the model has no uncertainty [33].

## 5. Conclusions

The research investigation of the five machine learning algorithms has been put into reality, and the digital twins model with its fundamental building block has been framed. PNN, SVM, KNN, NB, and RF were the five algorithms with the highest accuracy, with 91%, 87%, 86%, 83%, and 90%, respectively. Based on sensory information, the following conclusions were drawn:➢The DT model is constructed based on the findings of the validation tests, and it is then tested using data from various tool conditions. ➢The process requires data collection, the finetuning of the algorithms, and the modeling of digital twins based on the results. ➢Additionally, the DT model must depict the whole lifecycle of the instrument, including its design, manufacture, and maintenance. ➢Enhancing tool lifecycles may assist businesses in anticipating tool life and identifying the remaining valuable lifetime of the tool. This monitoring technique is quite successful. ➢To precisely trace the current level of tool wear, DT modeling is accomplished. Future product deployments using this technique are possible. ➢Numerous more machining procedures can also be performed using this technique.➢These digital twins can be presented as an application that can be used to monitor them online.

## Figures and Tables

**Figure 1 sensors-23-05431-f001:**
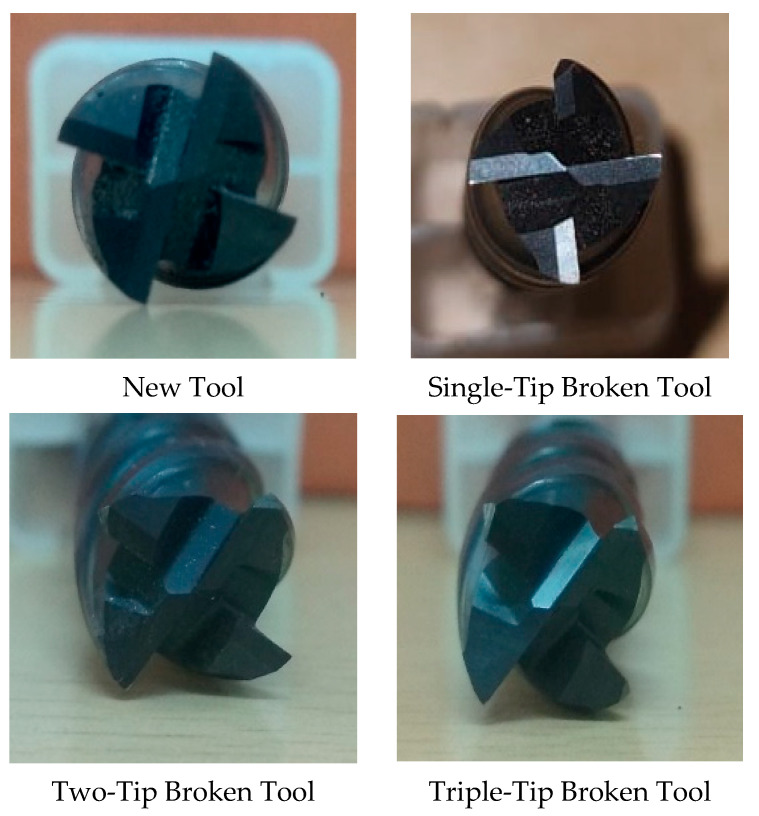
Various tool conditions.

**Figure 2 sensors-23-05431-f002:**
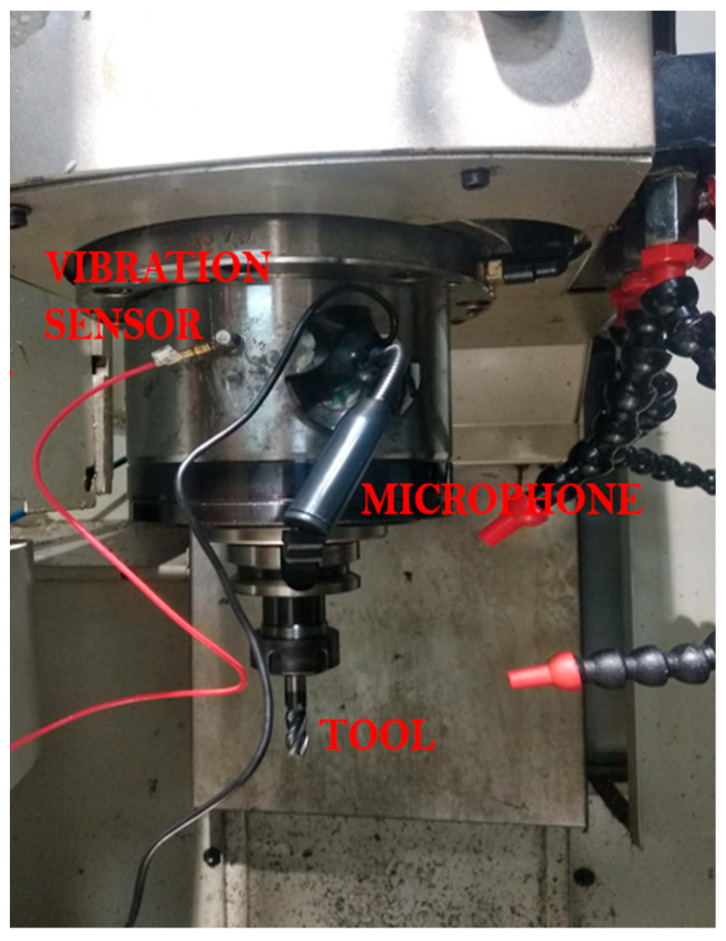
Experimental setup.

**Figure 3 sensors-23-05431-f003:**
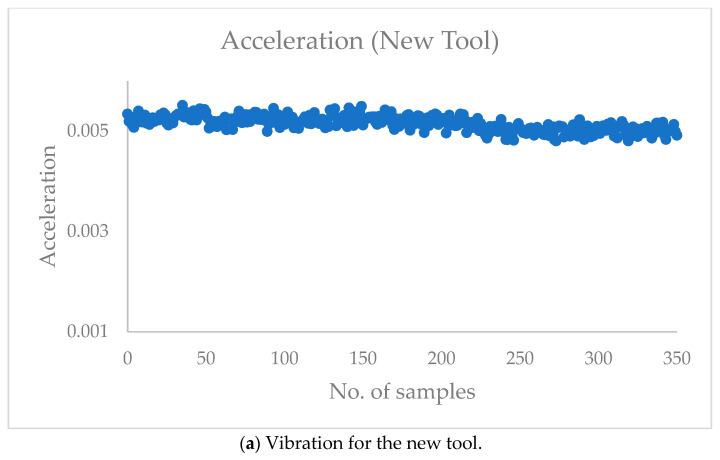
Vibration and sound signals for the new and four-tip broken tools.

**Figure 4 sensors-23-05431-f004:**
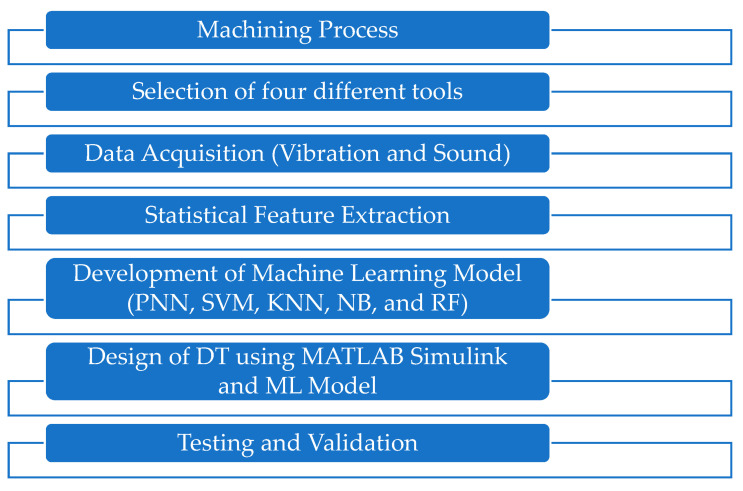
Proposed methodology.

**Figure 5 sensors-23-05431-f005:**
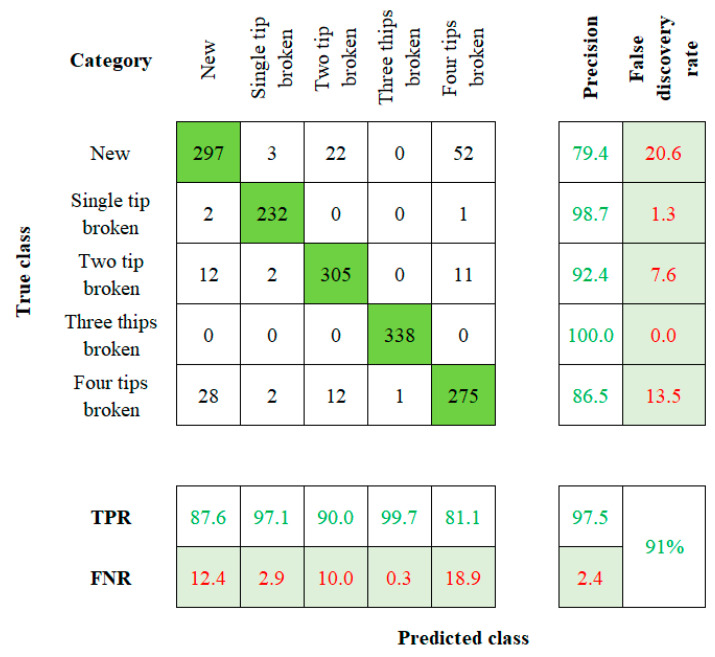
Confusion Matrix of PNN.

**Figure 6 sensors-23-05431-f006:**
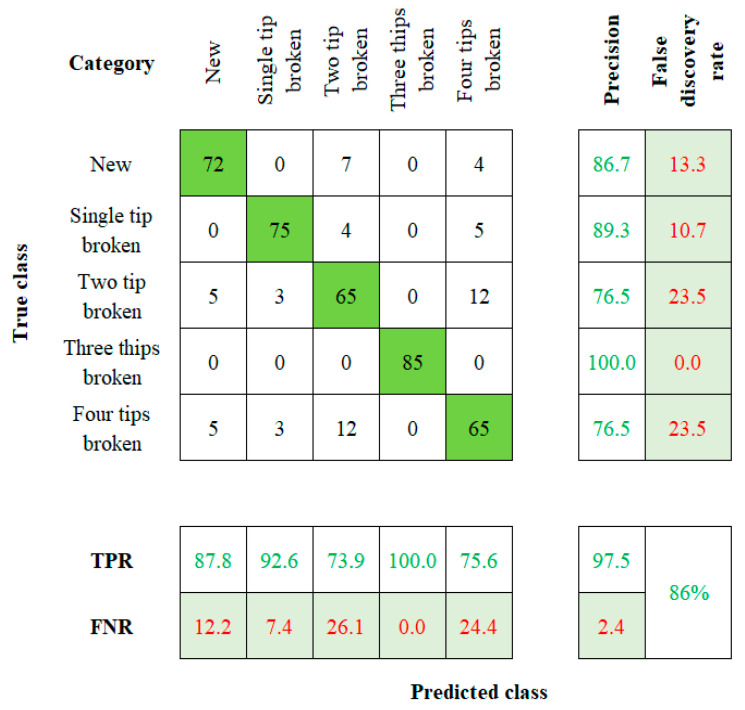
Confusion Matrix of KNN.

**Figure 7 sensors-23-05431-f007:**
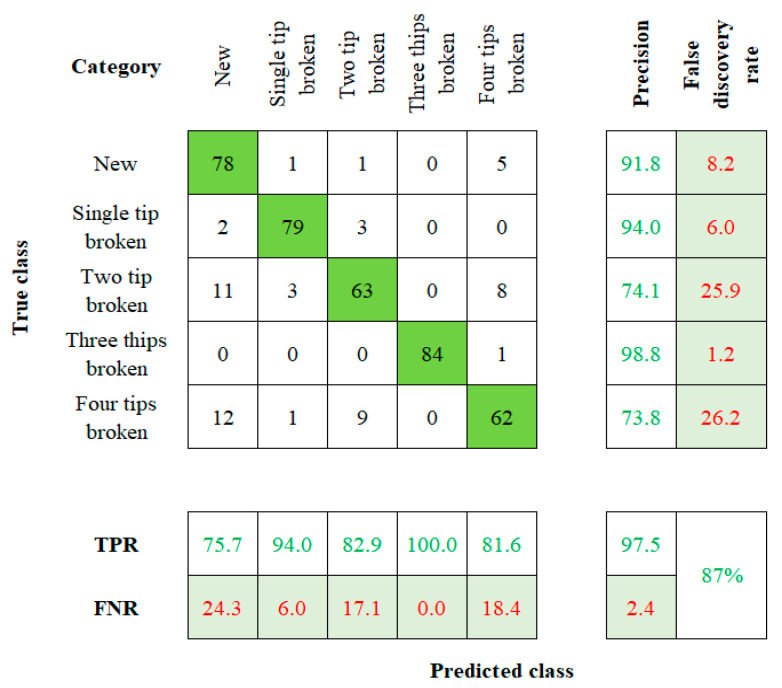
Confusion Matrix of SVM.

**Figure 8 sensors-23-05431-f008:**
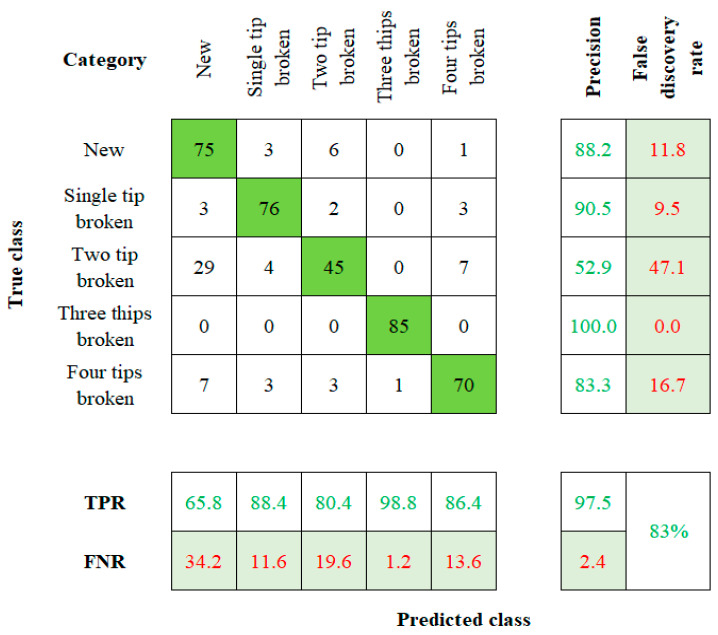
Confusion Matrix of NB.

**Figure 9 sensors-23-05431-f009:**
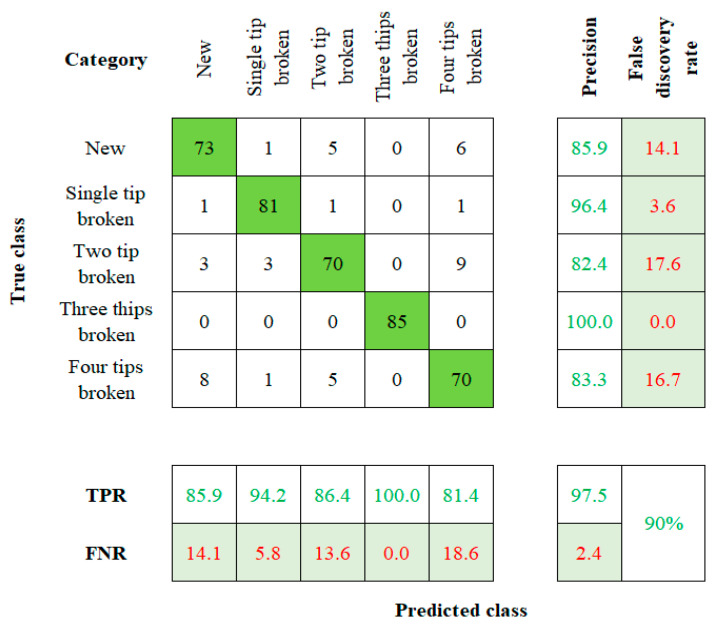
Confusion Matrix of RF.

**Figure 10 sensors-23-05431-f010:**
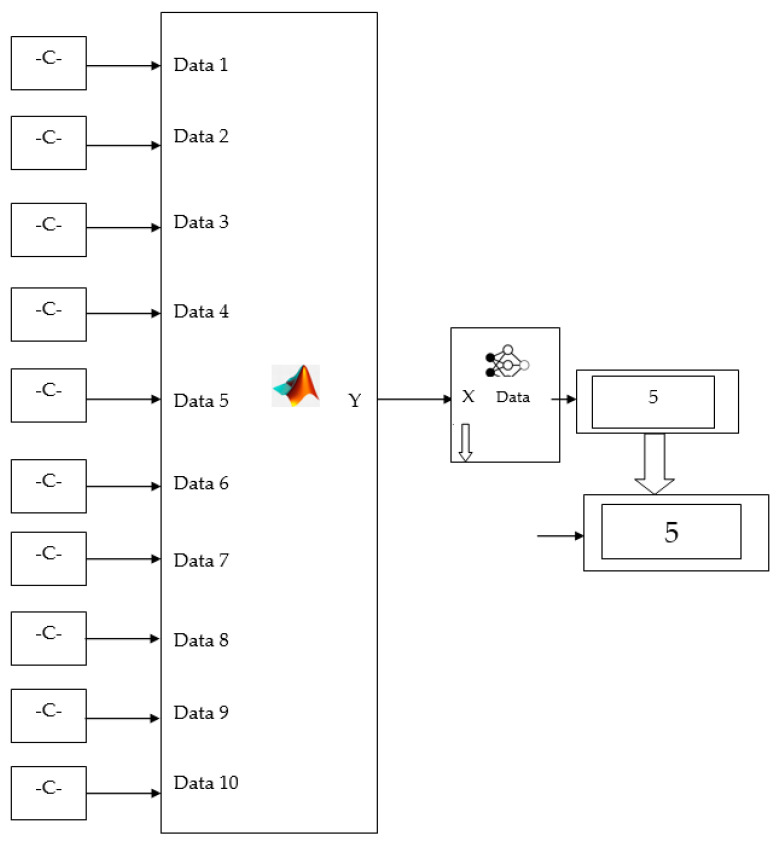
Test run of the DT model for the fifth condition of the tool.

**Table 1 sensors-23-05431-t001:** Machining parameters.

Machining Parameters	Values
Spindle speed	1800 rpm
Feed rate	250 mm/rev
Depth of cut	0.15 mm
Cutting condition	Wet/Commercial cutting fluid

**Table 2 sensors-23-05431-t002:** Prediction accuracy of tool condition using vibration signals.

Algorithms Used	Accuracy
PNN	91%
KNN	86%
SVM	87%
NB	83%
RF	90%

**Table 3 sensors-23-05431-t003:** Accuracy of TCMs using vibration signal in milling process from recent papers.

References	Algorithms Used	Accuracy
[12]	SVM	95.16%
	DT	94.35%
	KNN	89.86%
	NB	90.25%
	MLP	94.56%
[29]	GSVM	86.20%
	CSVM	85.70%
	KNN	85.70%
	DT	84.90%
	ANN	82.10%
[30]	SVM	98.70%
	KNN	93.70%
	DT	96.30%
[31]	DT	79.30%
	SVM-Cubic kernel	91.90%
	SVM	81.30%
[32]	CART	97%
	SVM	98%
	RF	94%
	KNN	99%

## Data Availability

Not applicable.

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
