# Peer review of "Digital Twin-Driven Tool Condition Monitoring for the Milling Process"

_sensors, 2023, doi:10.3390/s23125431_

Round 1

Reviewer 1 Report

This is a very good topic, but the organization of the author's research is somewhat beside the point.

1.In general, the model trained by measurement data is different from the twin model. However, the paper should elaborate the establishment of tool twin model from the point of view of mechanism. Measurement data should be used as the basis for revision of the model.

2. The 3D information twin model should be given.

3. Figure 1 and Figure 2 are not clear.

4. figure 3, figure 5 and figure 11 are not necessary; they do not clearly give the necessary twin model information.

Author Response

Sensors-2411503

Digital Twin-driven tool condition monitoring for the Milling process

The authors wish to thank the editor and reviewers for accepting our manuscript for further processing in your esteemed Journal "Sensors." The authors expressed their sincere thanks to the reviewers for finetuning this manuscript. We have taken utmost care to address all the comments raised by the reviewers with suitable justifications. The changes that we are made are highlighted in red color. We shall be happy if our revised manuscript is accepted for publication in view of our explanations in this reply and revision of the manuscript. The detailed information is given attached for your kind reference.

Reviewer 2 Report

This manuscript presents a technique dependent on digital twins to accomplish extraordinary accuracy in checking and anticipating tool conditions. Overall, the manuscript is well organized and innovative, and can be accepted after revision.

1.       How to select the best sensor placement. The recent advances should be referred: 10.1016/j.ymssp.2021.108386.

2.       What is the minimum data volume requirement for this method?

3.       Can non-contact measurement or wireless sensor systems be used?

4.       The robustness of the method under uncertainty and error should be explained. The recent progress on this topic should be reviewed including 10.1016/j.ress.2023.109382 and 10.1109/TAES.2023.3257777.

5.       Image clarity needs to be improved such as Fig. 11.

6.       How susceptible is the method to uncertainty? The author can refer the work: 10.1016/j.ast.2023.108155.

7.       The flowchart of Fig. 4 should be with more details.

well

Author Response

Sensors-2411503

Digital Twin-driven tool condition monitoring for the Milling process

The authors wish to thank the editor and reviewers for accepting our manuscript for further processing in your esteemed Journal "Sensors." The authors expressed their sincere thanks to the reviewers for finetuning this manuscript. We have taken utmost care to address all the comments raised by the reviewers with suitable justifications. The changes that we are made are highlighted in red color. We shall be happy if our revised manuscript is accepted for publication in view of our explanations in this reply and revision of the manuscript. The detailed information is attached below for your kind reference.

Reviewer 3 Report

Notes in the attachment.

Author Response

(The authors gave the same response as above.)

Round 2

Reviewer 1 Report

The paper is obviously improved based on the original manuscript, and the questions are supplemented and answered.

Author Response

Thank you sir.

Reviewer 3 Report

1. Please explain why the vibration signal was measured in the "z" direction? I know from experience that it is best to measure the signal in the feed direction.

2. How do the authors interpret the CTF occurrence? Please describe using the example of Figure 4. What measures in the time domain are used? No explanation in the text.

Author Response

The authors wish to thank the editor and reviewers for accepting our manuscript for further processing in your esteemed Journal "Sensors." The authors expressed their sincere thanks to the reviewers for finetuning this manuscript. We have taken utmost care to address all the comments raised by the reviewers with suitable justifications. The changes that we are made are highlighted in red color. We shall be happy if our revised manuscript is accepted for publication in view of our explanations in this reply and revision of the manuscript. The detailed information is attached below for your kind reference.
